# Hydrogen Nanometrology in Advanced Carbon Nanomaterial Electrodes

**DOI:** 10.3390/nano11051079

**Published:** 2021-04-22

**Authors:** Rui Lobo, Noe Alvarez, Vesselin Shanov

**Affiliations:** 1Laboratory of Nanophysics/Nanotechnology and Energy (N2E), Center of Technology and Systems (CTS-UNINOVA), NOVA School of Science & Technology, FCT-NOVA, Universidade NOVA de Lisboa, 2829-516 Caparica, Portugal; 2Department of Physics, NOVA School of Science & Technology, FCT-NOVA, Universidade NOVA de Lisboa, 2829-516 Caparica, Portugal; 3Department of Chemistry, University of Cincinnati, Cincinnati, OH 45221, USA; noe.alvarez@uc.edu; 4Department of Chemical and Environmental Engineering, University of Cincinnati, Cincinnati, OH 45221, USA; shanovvn@ucmail.uc.edu

**Keywords:** hydrogen storage, nanometrology, carbon nanotubes, graphene, nanotubes thread, desorption spectrometry

## Abstract

A comparative experimental study between advanced carbon nanostructured electrodes, in similar hydrogen uptake/desorption conditions, is investigated making use of the recent molecular beam-thermal desorption spectrometry. This technique is used for monitoring hydrogen uptake and release from different carbon electrocatalysts: 3D-graphene, single-walled carbon nanotube networks, multi-walled carbon nanotube networks, and carbon nanotube thread. It allows an accurate determination of the hydrogen mass absorbed in electrodes made from these materials, with significant enhancement in the signal-to-noise ratio for trace hydrogen avoiding recourse to ultra-high vacuum procedures. The hydrogen mass spectra account for the enhanced surface capability for hydrogen adsorption in the different types of electrode in similar uptake conditions, and confirm their enhanced hydrogen storage capacity, pointing to a great potential of carbon nanotube threads in replacing the heavier metals or metal alloys as hydrogen storage media.

## 1. Introduction

The development of a sustainable energy system is one of the most important challenges today and electrocatalytic hydrogen evolution is one of the most important ways to develop a sustainable energy system based on hydrogen technologies [1]. Together with suitable safe storage methods, this will pave the way for making this light fuel gas a sustainable energy source. Among electrochemical energy conversion advances, hydrogen energy is considered the most powerful candidate to alternate fossil energy due to its clean, renewable, and environmentally friendly properties and high energy density [2,3,4]. Despite the fact that in electrocatalytic water splitting the Pt cathode with near-zero overpotential is considered the most effective catalytic cathode, it has a high cost and platinum is a scarce resource [5]. With the purpose of seeking a low-cost and efficient catalytic cathode, many materials have been extensively explored by several authors.

Currently, many studies have been devoted to the use of hydrogen storage solids as active materials for application in fuel cells, and as a means of intermittency mitigation in energy supplies, with the aim of carbonic neutrality and renewable clean energy as urgent objectives. Activated carbons are ineffective in hydrogen storage systems because only a small fraction of pores in their pore-size distributions are small enough to interact strongly with hydrogen molecules at the gas phase. For small-sized systems, surface- or interface-related sites assume crucial importance and can change the overall solubility of hydrogen. Thus, for improving the kinetic characteristics of hydrogen storage in materials, great attention has been paid to thermal desorption studies and calibration [6,7,8]. Thermal desorption spectrometry (TDS) assures a good accuracy in monitoring hydrogen absorbed in solid materials [9], and it is suitable for monitoring hydrogen evolution following the application of a thermal ramp, in contrast to the determination of static sorption isotherms. The use of a mass spectrometer allows evolved gas species to be identified, whereas the other methods are non-selective. The binding energy of adsorbed molecules varies with the nature of the adsorbate/surface materials, and desorption temperature gives information on the binding energy [10,11].

An additional goal consists of looking for low weight and high performance hydrogen active storage media made from carbon nanomaterial assemblies. Several carbon nanomaterial assembly-based electrodes are used in this work for comparing their relative performance. With this purpose in mind, the recent molecular-beam thermal desorption spectrometry (MB-TDS) is used in this work [12]. This technique is a variant of thermal desorption spectrometry which has been developed to detect the hydrogen release by the lowest amounts of solid samples, with several advantages [12,13,14,15,16].

## 2. Materials and Methods

A tuned desorption spectrum for hydrogen typically displays peaks caused by desorption with an origin on different adsorbed phases, and the total desorbed quantity of hydrogen is determined from the integral of the mass spectrometer signal, providing it can be adequately calibrated [12]:(1)$$n_{T}=\int_{t_{0}}^{t_{1}}n\left(t\right)\mathrm{dt}$$
where n_T_ represents the number of moles of desorbed hydrogen and n(t) the molar hydrogen desorption rate as a function of time.

Within the range of temperatures used in this work (below 400 K) hydrogen desorption from chemisorbed sites in the material is not detected and so one can simplify the non-equilibrium kinetics of the global desorption process by the following three elementary steps:(i)(C–H)_ads_ → H(lat)(ii)H(lat) → H(s)(iii)H(s) + H(s) → H_2_(g)where H(s) are hydrogen atoms on the surface of the material grains or nano-regions of the material, H(lat) are hydrogen atoms inside the material lattice but outside the chemisorption centers, and (C–H)_ads_ are the physisorbed hydrogen atoms on active sites. The second stage is diffusion-limited, and from the condition of maximum desorption at *T* = *T_m_*, one gets:(2)$$\ln\left(\frac{T_{m}^{2}}{\beta }\right)\approx \frac{E_{a}^{dif}}{k_{B}T\left(T_{m}\right)}$$
where *E_a_^dif^* is the diffusion activation energy, *k_B_* the Boltzmann constant, and *β* the heating rate.

The activation energy of desorption *E_d_* can then be determined from the linear dependence of ln(*β*/*T_m_*^2^) on 1/*T_m_* and from these considerations, it is then possible to confirm or not, if desorption is a diffusion-controlled process [17,18].

Pumping out the heated oven (with the sample inside), and considering that desorption activation energy, *E_d_* it is possible to show that desorption activation energy can be obtained from the experimental MB-TDS desorption curves making use of the equation [19]:(3)$$\ln\left(\frac{\beta }{T_{m}^{2}}\right)=-\frac{E_{d}}{RT_{m}}+\ln\;\frac{RB}{E_{d}}$$
where *B* stands for the Arrhenius pre-exponential factor, and *R* is the universal gas constant.

Molecular-beam thermal desorption mass spectrometry is applied to the determination of electrochemical hydrogen uptake and release by the samples. The MB-TDS apparatus has already been described elsewhere [12] and is merely schematically displayed in Figure 1, where a composite molecular beam of known intensity is produced from the degassing solid sample at a certain temperature inside the oven.

By tuning the mass spectrometer to the hydrogen gas, one can monitor its evolution in time, thus measuring the amount of hydrogen desorbed from the sample. The effusion beam can be geometrically defined as well as its fraction detected by the quadrupole mass spectrometer (QMS) located in the forward direction (Figure 1).

Values of the order of 1 °C min^−1^ have typically been used for the heating rate of each sample. The hydrogen partial pressure background variation can be described by a temporal decay [12,13].

By subtracting the residual hydrogen gas background (measured without the beam) from the total amount of hydrogen impinging in the quadrupole spectrometer, one can compute the real amount of hydrogen coming from the sample [12,13]. The ratio of the hydrogen background partial pressure variations, for the situations of heating-on and heating-off, lead to the simple following expression [12]:(4)$$\frac{N_{0}+N_{b}}{N_{0}}=1+\frac{A_{b}}{A_{0}}$$

*N* and *A* stand, respectively, for the number of hydrogen molecules and the area under the desorption curve, with the subscripts “0” and “*b*” assigned respectively to the situations “without the beam” and “with the beam”.

Before introducing a sample in the oven of the MB-TDS apparatus, it needs to be submitted to hydrogen uptake, and this is done by electrochemical hydrogen charging. This procedure requires a previously experimental preparation of cathodes, which is summarized here for different carbon nanomaterials.

(i)Graphene

Graphene structures have received considerable interest due to their outstanding properties such as interconnected porous structures, enormous specific surface area, good mechanical stability, and flexibility to tailor their surface chemistry [20,21,22].

A 3D-graphene in the form of a pellet, possesses a monolithic structure with high electrical conductivity (148 S cm^−1^), well-controlled mesopore size (~2 nm), and good electromechanical properties. Details about the synthesis and characterization of 3D graphene have been published by our group elsewhere [23,24,25,26]. In brief, 3D graphene pellet is synthesized by chemical vapor deposition (CVD), using nickel powder as a catalyst, and it can be further processed into a graphene paper by pressing. It possesses electrical conductivity of above 1100 S cm^−1^ and exhibits breaking stress of about 20 MPa.

Since Ni is catalytic, the etching with HCl was active for one hour, which is believed to be time enough to assure a complete depletion of Ni contamination. There was no determination of the Ni that remained.

The morphology of graphene with high surface roughness could contribute to a good adhesion with any substrate of practical interest due to the existence of van der Waals forces. Figure 2a displays a scanning electron microscopy (SEM) image of the 3D graphene pellet and Figure 2b shows Raman spectra of the same sample conducted at different spots.

After etching the Ni catalyst with HCl acid, the graphene pellet obtained was transferred onto a Kapton (polyimide) film and dried at 40 °C. A good adhesion between graphene and Kapton substrate was created due to van der Waals forces and the unique morphology of the 3D graphene. The graphene pellet with the dimension 5 mm × 5 mm was connected to a silver wire using silver conductive epoxy, which was then coated with non-conductive epoxy. After drying overnight at room temperature, the working electrode was ready for use.

(ii)Single-walled carbon nanotube networks

Uniform thin films of single-walled carbon nanotube (SWCNT) networks of varying densities have been fabricated at room temperature by vacuum filtration. A dilute suspension of purified rice nanotubes in chloroform is sonicated, and then vacuum filtered through an alumina membrane (Whatman, Maidstone, UK, 20 nm pore size) in a short time (few seconds). As the solvent passes through the pores, the nanotubes are trapped on the surface of the filter, forming an interconnected mesh. Its density (nanotubes/area) can be selected by controlling the volume of the dilute suspension filtered through the membrane. The fast vacuum filtering process prevents tube flocculation, and then one proceeds with the transfer of the film to another surface by membrane dissolution. The film can be made free-standing over an aperture (25 mm^2^) by making the transfer to a Teflon substrate with a hole, over which the film is laid before membrane dissolution in sodium hydroxide aqueous solution. Measurements of the sheet conductance have shown that it increases with the network density. This is expected since through percolation a significant number of both metallic and semiconducting nanotubes operate as electrical conducting pathways. For an SWNT (single-walled carbon nanotube) sheet made from 20 mL of a solution of carbon nanotubes in chloroform with the concentration of 0.2 mg L^−1^ the measured DC electrical conductance was about 7 × 10^−6^ S cm^−2^. Finally, the free-standing film obtained is transferred on to a Kapton film and dried which is put in adhesion contact with it through the hole. Its atomic force microscopic topographical image is shown in Figure 3. The SWNTs network with the dimension 5 mm × 5 mm was connected to a silver wire using silver conductive epoxy, which was then coated with non-conductive epoxy. After drying at room temperature, the electrode was ready for use.

(iii)Multi-walled carbon nanotube networks

Vertically aligned multi-walled carbon nanotube (MWCNT) arrays were synthesized using CVD and were then removed from the catalyst substrate in a procedure described elsewhere [27] and transferred to a glass substrate. With a small roller of Teflon, the carbon nanotube (CNT) carpet is smashed against the substrate surface and converted into a free-standing film which is then submitted to the analogous previous procedures (i and ii) to build the electrode.

(iv)Carbon nanotube thread

MWNTs (multiple-walled carbon nanotubes) thread dry spinning starts with the synthesis of vertically aligned CNTs which have the ability to assemble themselves into films or ribbons, and subsequent threads [28]. Catalyst film thickness and CNT synthesis parameters have been studied [27]. The as-synthesized CNT arrays are completely detached from their substrates during synthesis and they are fully spinnable which allows monitoring the diameter of CNT threads [28]. Different twist angles from the same array width were obtained by varying the draw and rotational speed allowing a thread to be prepared by this method [29]. Figure 4 displays SEM images of the material.

A CNT pristine thread with 35 μm diameter and 2 cm long was connected to a copper wire using silver conductive epoxy. After that, it was coated with a polystyrene solution (15 wt% in toluene) and air-dried at 50 °C. Following, the CNT thread was aspirated into a glass capillary and the end of the glass capillary was sealed with a hot glue gun. Last, the polystyrene-coated CNT thread electrode was cut off at the end with a sharp blade. In this way, only the end of the CNT thread was exposed.

Usually, electrochemical hydrogen charging and discharging curves are recorded (by galvanostatic or voltammetry techniques) in order to estimate the electrochemical hydrogen storage capacity [30]. In this work, the hydrogen uptake in the nanostructured carbon electrodes was undertaken in a galvanostatic way, and the release of hydrogen was measured with the MB-TDS technique.

During electro-reduction (i.e., the process of insertion into nanoporous carbon), hydrogen could be stored by cathodic decomposition of water in both NaOH and H_2_SO_4_ aqueous solutions. Applying a negative polarization to the carbon electrode, hydrated Na^+^ and H_3_O^+^ cations are adsorbed, respectively, in the alkaline and acidic medium, forming the well-known electric double layer. Once the electrode potential becomes lower than the equilibrium redox potential, hydrogen in the zero-oxidation state is formed by the reduction of water in the case of alkaline solution (or of hydronium ions H_3_O^+^ in the acidic medium). In the next step, hydrogen is expected to be physically adsorbed onto the carbon surface giving rise to H_ad_ [31].

It is worth noting that typical impurities of NaOH and H_2_SO_4_ are catalytic, and so we used, respectively, NaOH reagent grade, ≥98%, pellets (anhydrous), and H_2_SO_4_ 99.999%.

The total amount of hydrogen adsorbed, (i.e., storage capacity), depends essentially on the kinetics of hydrogen diffusion and incorporation into the nanopores. At low overvoltage, the diffusion of H_ad_ proceeds slower than the reduction step of water or H_3_O^+^, then H_ad_ participated in chemical (Tafel) or electrochemical (Heyrovsky) recombination reactions, which result in the evolution of di-hydrogen. This latter case takes place with an H_2_SO_4_ medium, and it has been shown that hydrogen is poorly adsorbed, easily giving rise to molecular hydrogen evolution. In contrast, with the NaOH electrolyte, due to the high value of polarization, carbon demonstrates a noticeable hydrogen capacity. Hence, the electro-reduction of water in the basic medium allows higher hydrogen pressure to be reached than in the conventional gas phase technique [31]. Hydrogen produced from electrolysis during charging of the cell partly entered the working electrode and the rest was released as gas. The operation is interrupted after a rapid generation of hydrogen gas bubbles is observed, suggesting that storage was complete [30].

Electrolysis can be used practically to charge the material. Electrochemical hydrogen charging and discharging curves are recorded (by galvanostatic or voltammetry techniques) in order to estimate the electrochemical hydrogen storage capacity. In this work, the hydrogen uptake in the nanostructured carbon electrodes was done in a galvanostatic way, and the release of hydrogen was measured with the MB-TDS technique. A pure platinum anode was used for charging (at room temperature and atmospheric pressure) the working cathodes (already mentioned) of the electrolytic cell, through one molar NaOH electrolyte solution. All the samples were submitted to an identical constant applied voltage of 1.0 V in one hour. This was done at a constant applied voltage only after a significant increase in the electrolyte resistance had occurred. The cell contained an aqueous electrolyte of NaOH at a concentration of 1 M, and an electric current of 133 mA was imposed for 60 min. 

This galvanostatic procedure is common to the following four cathodes: graphene, single-walled and multi-walled carbon nanotube networks, and carbon nanotube thread. After the electrochemical charging, each sample was dried in a desiccator under a vacuum (Appendix A). 

Regarding sample (iv), coaxial cylindrical stainless steel was used as the anode in a similar electrolyte solution and similar hydrogen electrochemical charging conditions.

The samples were weighed in a Kern ABT-101 analytical balance before and after the electrochemical hydrogen uptake. The relative weight increases were: 9% for the MWNT thread; 8% for Pd; 6% for graphene; 4% for the SWNTs network and 2% for the MWNTs network. The differences in weight are in the order of a few units of milligrams.

Each of the hydrogenated samples was then submitted to the already described MB-TDS procedure, allowing interpretation of desorption spectra in a similar way to the well-known temperature programmed desorption spectroscopy [9,32,33,34]. By subtracting the residual hydrogen background pressure, measured without beam taken in similar experimental conditions, one can compute the total amount of hydrogen contained in the samples, from the area of the respective MB-TDS experimental curves (Figure 5) [34].

This way one can obtain the ratio *E_d_*/*R* and from that by subtracting the corresponding background pressure, one can obtain the area *A_b_* below them. This area is related to the number of hydrogen molecules *N_b_* detected with the QMS, by the expression [34]
(5)$$N_{0}+N_{b}=\left(A_{0}+A_{b}\right)\frac{N_{A}S}{RT}$$
where *S* is the pumping speed and *N_A_* the Avogadro number.

To estimate the total amount of hydrogen desorbed from the samples, the number of hydrogen molecules still needs to be corrected with the geometrical fraction of the hydrogen beam effectively “seen” by the QMS detector and the ionization efficiency of this detector as well [34]. An efficiency of about 70% is expected, given the QMS catalog technical specifications [12]. Also, from the geometrical configuration, one can estimate a fraction of 20% of the beam that effectively impinges on the QMS detector. From the combination of these two attenuation effects, a reduction factor in the order of 10 in the molecular flux impinging the detector is expected [12]. The hydrogen masses detected can then be computed, and not surprisingly, they are only in agreement with the relative weight increased values (previously recorded in the experimental part), if they are multiplied by a factor of 10, which is exactly the experimental attenuation factor already mentioned. The total amount of hydrogen content for each of the samples could then be computed from the MB-TDS spectrum, which offers an alternative way to avoid misleading weight results in the case of minimal amounts of hydrogen uptake [34].

## 3. Results and Discussion

From the relative corrected area of the spectra given in Figure 5, it is found that they reasonably agree with the expected relative weight increases already mentioned in the experimental part:MWNT thread = 9% → Graphene = 6% → SWNT network = 4% → MWNT network = 2%

The wt% of hydrogen absorbed in carbon nanostructures is likely to be proportional to their respective specific surface areas [35], and in fact, the above sequential order seems to be explained by this argument, as from all the carbonaceous samples, the carbon nanotube thread and the graphene pellet have the higher values of specific surface areas. This has been observed for materials with a surface area lower than 1000 m^2^/g [35]. However, for adsorbents with high surface area, some theoretical studies point to a relevant influence of the pore size distribution [36].

These advanced carbon nanostructures are very promising for hydrogen storage technological applications since they are much lighter than alternative metals (as palladium) or metal hydrides in metal alloys.

Recording MB-TDS spectra with different scan rates (0.5 K min^−1^, 1.0 K min^−1^ and 2.0 K min^−1^), and fitting linear plots to the experimental points (*β_i_*, *T_mi_*), one could find the corresponding desorption energies [32,33,34]:Graphene = 5.4 kJ/mol → MWNT network = 18.3 kJ/mol → SWNT network = 20.2 kJ/mol → MWNT thread = 22.1 kJ/mol

Figure 6 and Figure 7, respectively, display, those spectra and corresponding plots, for the case of the MWNT thread. Carbon materials usually display a high cycle life of hydrogen uptake/release and are lighter and mostly environmentally friendly, in comparison with pure metals or metal hydrides. The electrochemical hydrogen storage capacity of carbon materials exceeding 1.5 wt% at ambient conditions allows them to be viewed as a potential anode material in replacement of some pure metals or metallic alloys [37].

The hydrogen storage in CNTs is the result of the combined action of physisorption and chemisorption. The hydrogen storage capacity of carbon materials mainly depends upon surface area, which is affected by micropore size distribution that counts for the presence of narrow micropores. Based on theoretical studies, some authors have found a strong structural dependence on hydrogen adsorption in carbon-based materials (especially CNT), and defects affect their adsorption capacity.

Many researchers have carried out experimental research and theoretical analysis on hydrogen storage in carbon nanotubes (CNTs), but the results are very inconsistent [38]. Hydrogen electro-sorption on carbon materials is a complex process, involving the overlapping of hydrogen ad-atoms formation and storage. If carried out under the same electrochemical conditions, this combined process depends on the porous structure and surface chemistry of the sorbent material as well as its electrical properties. The highest efficiency is achieved if the carbon materials are very amorphous and have a hierarchical micro/mesopore structure [39]. One can assume that electrochemical charging consists of two steps: (1) basic charging, which proceeds with increased H_ad_ coverage of the outer surface of the porous material; and (2) a surcharge, consisting in continuous saturation of less accessible narrow micropores with hydrogen when H coverage of the outer surface is high enough [37].

The electrochemical charge-discharge mechanism in SWCNT paper electrodes is known to be controlled by a proton diffusion process, and somewhere in between a physical process (as in pure carbon nanotubes) and a chemical process (as in metal hydride electrodes). It consists of a charge transfer reaction (reduction/oxidation) and a diffusion step (diffusion) [40].

Electrochemical studies have also revealed that oriented mesoporous carbon shows better electrochemical storage of hydrogen compared to ordinary activated carbon [41]. For large diameter single-walled carbon nanotubes (SWCNTs), a hydrogen storage capacity of 4.2% weight (i.e., hydrogen to carbon atom ratio of 0.52), has been achieved (at room temperature and under the pressure of 10 MPa), and 3.3% by weight could be released at ambient pressure and room temperature, while the release of the residual stored hydrogen (0.9 wt%) required some heating of the sample [42]. The following experimentally measured hydrogen storage capacities of carbon nanotubes have been reported: SWCNT (8 wt%), Li-doped MCWNT (20 wt%), Well-aligned MCWNT (3 wt%). The accuracy of the method used for the determination of hydrogen electrosorption ability is limited mainly by the accuracy of sample mass determination [43].

Temperature programmed desorption data for hydrogen absorbed in SWNTs using gas-phase loading revealed activated desorption energy of about 19.6 kJ mol^−1^ [44]. This value is close to that obtained in this work for the SWNT network which confirms the absorption performance of SWNT assemblies by two different techniques.

In the case of physical absorption, the desorption activation energy corresponds to the heat of adsorption, and such observed highly bound physisorbed di-hydrogen is consistent with adsorption within the SWNTs cavities, indicating that the inner parts of the tube become accessible after heating under vacuum [44]. The value of the activated desorption energy found for the SWNT network sample is very close to the value obtained by using hydrogenated SWNTs in the gas phase.

Since the hydrogen evolution reaction would have a large overpotential on the basal plane of graphene, the reaction most likely occurs either on edge functional groups or on metallic impurities, so it will be necessary to characterize the samples much more thoroughly, with this thought in mind. Also, as the hydrogen evolution reaction is facile on Pt it is dangerous to use a Pt anode because, during electrolysis, Pt dissolves and goes into solution, redepositing on the cathode and possibly enhancing the catalysis. In future works, it will be recommended to check the effect of using a non-catalytic anode material.

Hydrogen can be physisorbed in carbon nanotubes bundles on various sites such as external wall surface, grooves, and interstitial channels. Therefore, it can have a large energy density (as required for mobile applications). It is also known that by tuning the adsorption conditions hydrogen can be either chemisorbed or physisorbed on carbon nanotubes. The adsorption of atomic hydrogen is highly unfavorable on the basal plane of graphene and would require specific sites, probably metallic. However, in the MB-TDS experiments, our mass spectrometer is unable to distinguish between adsorbed molecular and atomic hydrogen. For that, a QMS with improved mass resolution will be needed.

## 4. Conclusions

Results reported here stand for the analytical determination of absorbed/adsorbed hydrogen using a specific type of thermal desorption spectroscopy in conjunction with mass spectrometry (MB-TDS). This technique is revealed to be a powerful procedure for monitoring, in real-time and in situ, the hydrogen release from different advanced carbon nano-electrocatalysts, and for a comparative quantitative analysis of this trace element among them (the amounts are below the detection limit of a microbalance).

The hydrogen storage in CNTs is the result of the combined action of physisorption and chemisorption. It has been demonstrated that the maximal degree of nanotube hydrogenation depends on the nanotube diameter, and for the diameter values around 2.0 nm, nanotube-hydrogen complexes with close to 100% hydrogenation exist and are stable at room temperature. This means that specific carbon nanotubes can have a hydrogen storage capacity of more than 7 wt% through the formation of reversible C–H bonds.

The results point to the use of inter-tube sites in carbon nanostructures for storing hydrogen by a physisorption mechanism. Temperatures of hydrogen desorption as high as about 350 K has been achieved, possibly due to the availability of ‘sub-nanometer’ sized spaces [45].

This study is one step further in comparing advanced carbon nanostructured electrodes for similar hydrogen uptake/desorption conditions and confirms the expectations about their enhanced hydrogen storage capacity.

The MWNT thread displays the largest desorption energy among them and possesses a set of suitable properties for making it an excellent candidate for electrocatalyst for hydrogen evolution.

The results of this investigation on the H_2_-storage ability of various carbon-based nanomaterials indicate that MWNT thread material has the highest wt% of H_2_ absorbed as high as 9% compared to 6%, 4%, and 2% for graphene, SWNT network, and MWNT network, respectively. This confirms that the hydrogen storage capacity of carbon materials depends on the surface area. As a suggestion for further work to be pursued, it will be interesting to establish the relationship of the H_2_ capacity against the surface areas as well as pore volumes of the corresponding materials. The availability of BET data on surface area and pore volumes for all the studied materials eventually will illustrate such correlation, albeit the N_2_-adsorption-desorption isotherm data are not able to directly reflect the H_2_ storage ability of the materials (N_2_ isotherm can provide a positive trend for the capacity ability toward hydrogen molecules). Future studies on surface modification and composite structure are expected to be carried out, in the search to improve the electrocatalytic performance by overcoming the limitations of active sites, intrinsic catalytic activity, and inter-yard conductivity. In-depth research on these issues can provide a clue for improvement of the efficiency in electrocatalytic hydrogen evolution and a deep insight into the catalytic mechanism.

## Figures and Tables

**Figure 1 nanomaterials-11-01079-f001:**
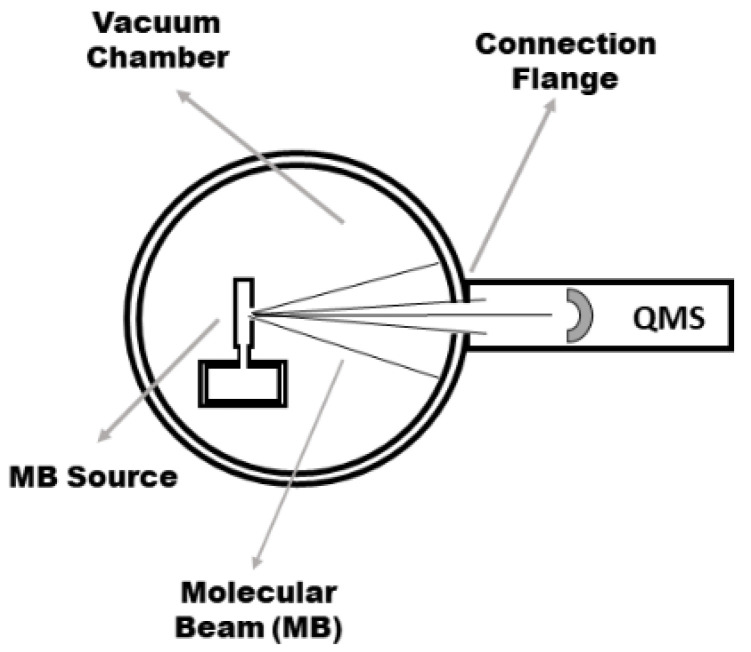
Top view of the molecular-beam thermal desorption spectrometry (MB-TDS) configuration.

**Figure 2 nanomaterials-11-01079-f002:**
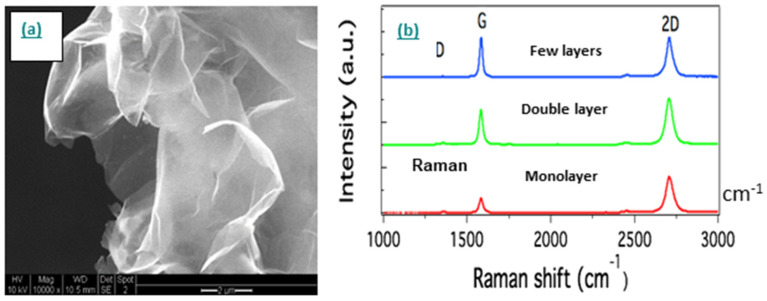
(**a**) Scanning electron microscope (SEM) of 3D graphene pellet; (**b**) Raman spectroscopy of 3D graphene pellet conducted at altered spots on the sample, revealing a different number of graphene layers within the probed flakes.

**Figure 3 nanomaterials-11-01079-f003:**
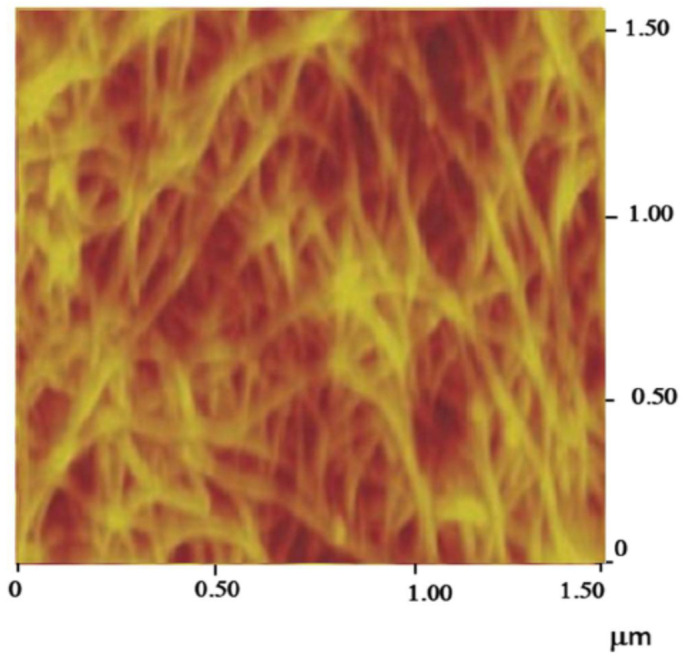
Atomic force microscopy (AFM) image of SWCNT network transferred to a film surface.

**Figure 4 nanomaterials-11-01079-f004:**
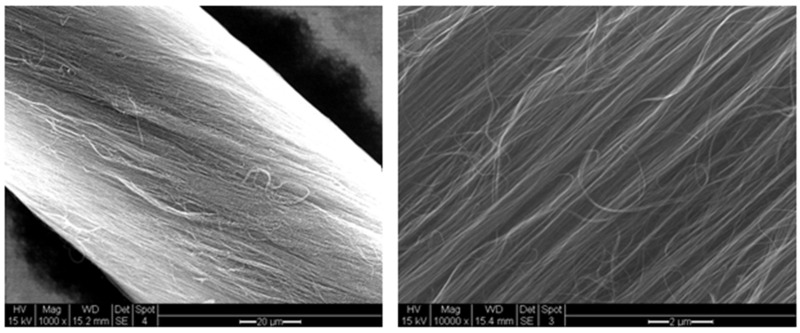
SEM images of carbon nanotube thread.

**Figure 5 nanomaterials-11-01079-f005:**
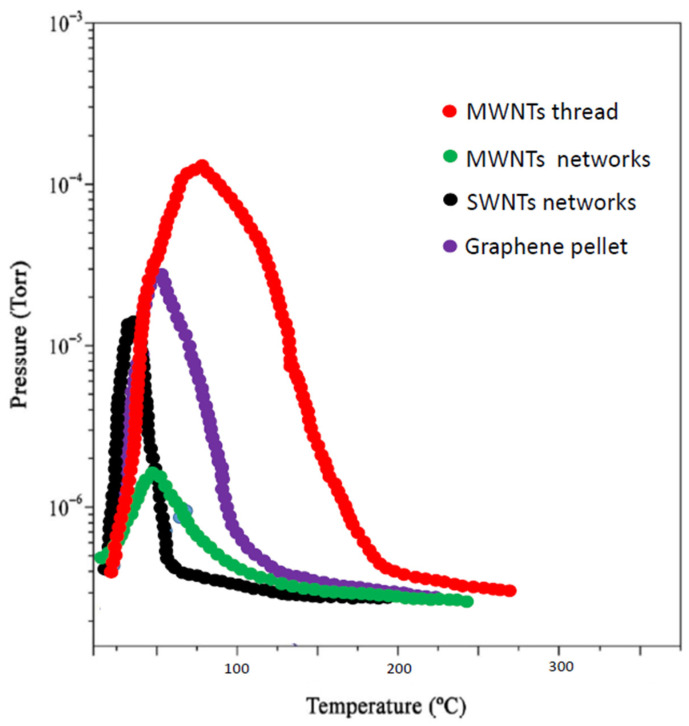
MB-TDS spectra were taken at a heating rate of 1 °C/min.

**Figure 6 nanomaterials-11-01079-f006:**
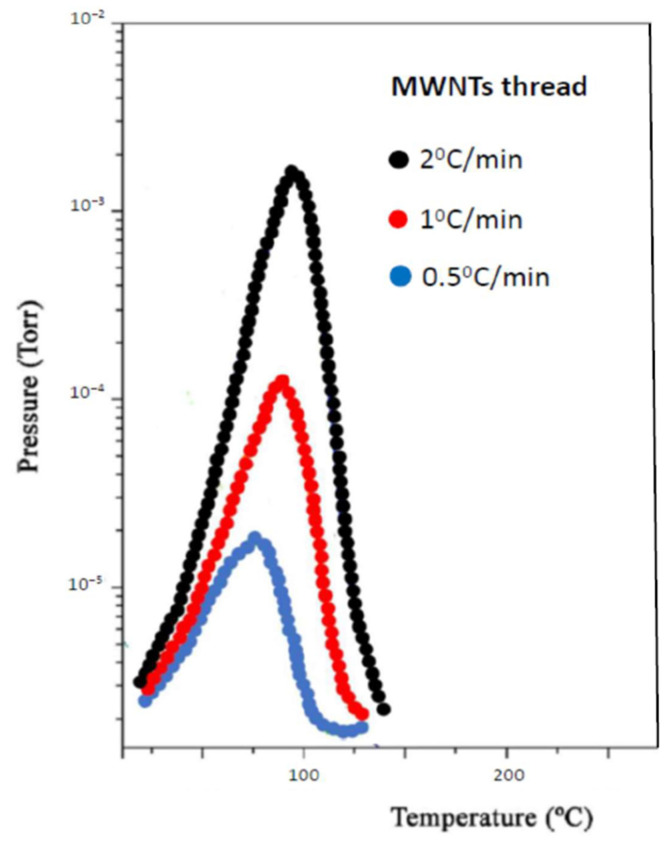
MB-TDS spectra of hydrogenated MWNT thread, taken at three different heating rates.

**Figure 7 nanomaterials-11-01079-f007:**
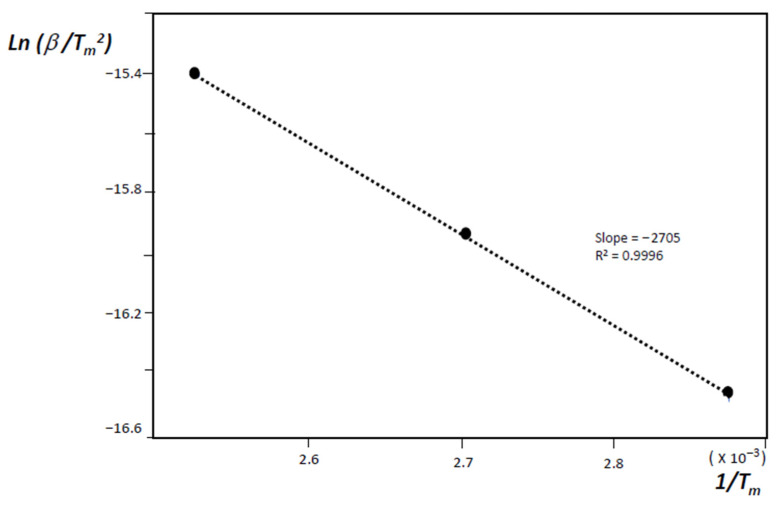
Fitting of a linear plot to the experimental points (*β_i_*,*T_mi_*) obtained for hydrogenated MWNT thread.

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
