# Peer review of "Hydrogen Nanometrology in Advanced Carbon Nanomaterial Electrodes"

_nanomaterials, 2021, doi:10.3390/nano11051079_

Round 1
Reviewer 1 Report
After reading the manuscript entitled "Hydrogen nanometrology in advanced carbon nanomaterial electrodes" by Lobo et al. I believe that the work is interesting and prepared correctly according to the art of scientific work. However, it requires a little enrichment.
My general remarks are:
- the organisation and writing of the paper are of good quality,
- the presentation of the materials and methods is overall clear and easy to follow apart from some missing details,
- the discussion of the results is clear and most conclusions are sound and relevant. A few questionable assumptions should be revised, though,
- the paper opens up interesting perspectives for further improvement.
Hereunder are my detailed comments:
1. Page 2, line 78. Replace “b” by “beta”.
2. Page 2, line 80, ln(Tm2/ ). Problem with this formula – see Fig. 7 (y-axis) and Eq. (3). ln(Tm2/ ) ----> ln(beta/Tm2).
3. Page 2, line 84, [20]. Wrong literature reference. I can't find Eq. (3) in [20].
4. Page 2, line 87, “Molecular-beam thermal desorption mass spectrometry (MB-TDS)”. Duplicated abbreviation – see line 57.
5. Page 3, line 97. forward direction. ---> forward direction (Fig. 1).
6. Page 5, line 176 and others. There is no clear end of paragraph (iv). Maybe insert a blank line?
7. Page 7, lines 243 and 244, “The wt% of hydrogen absorbed in carbon nanostructures is likely to be proportional to their respective specific surface areas”. I cannot agree with this statement. It is true for materials with the surface area lower than 1000 m2/g – see Fig. 2 [35]. For adsorbents with high surface area see Kowalczyk, Hydrogen storage in nanoporous carbon materials: myth and facts, Phys. Chem. Chem. Phys., 2007, 9, 1786-1792, https://doi.org/10.1039/B618747A.
8. Pages 7 and 8, Figs. 5 and 6. Please replace points by line.
9. Page 7, line 274, Had. “ad” – subscript.
10. In the case of carbon nanotubes, will the diameter of the carbon nanotubes affect the results?
11. Page 11, line 407, [36]. Krzysztof. J. ---> Jurewicz K.
Author Response
Green : Reviewer sugestions have been taken into account in the revised manuscript. Yellow: Comments and rebutal to the reviewer suggestions. Blue: Additional text included as responses to the reviewer suggestions.

Reviewer 2 Report
The authors have reported results for the analytical determination of absorbed/adsorbed hydrogen using a specific type of thermal desorption spectroscopy in conjunction with mass spectrometry. The results seem to be interesting and could be published with some additional clarification and inclusion of additional results and possibly additional measurements.
Generally speaking, this research area seems to be in its infancy and will benefit from the establishment of a base of detailed results. Here are some of the points that will require consideration or attention:
1. Since the hydrogen evolution reaction (HER) would have a large overpotential on the basal plane of graphene, the reaction most likely occurs either on edge functional groups or on metallic impurities, so it will be necessary to characterize the samples much more thoroughly, with this thought in mind.
2. As the authors point out, the HER is facile on Pt. Therefore, it is exceedingly dangerous to use a Pt anode, because, during electrolysis, Pt dissolves and goes into solution, redepositing on the cathode and possibly enhancing the catalysis. It is necessary to check the effect of using a non-catalytic anode material.
3. Since the readership is somewhat general, it is necessary to comment on whether electrolysis can be used practically to charge the material, or is it somewhat academic?
4. Practically, evolving H2 in a porous electrode can make said electrode expand, lose conductivity and even explode.
5. The discussion of both atomic and molecular hydrogen is confusing. Again, the adsorption of atomic hydrogen is highly unfavorable on the basal plane of graphene and would require specific sites, probably metallic.
Specific comments follow:
Line 61. “Tunned” —> “tuned”
Line 119. “Details published elsewhere” Here and in several other places, it is necessary to insert a brief few words of explanation.
Line 127. How much Ni still remains in the material. As the authors are aware, Ni is catalytic.
Line 134. SWNT —> SWCNT
Line 156. MWNT —> MWCNT
Line 158. CNT is used without definition
Line 176. Electrochemical charging curves are mentioned but not shown; these could be included in Supplementary Material.
Line 204. Source and purity of NaOH, H2SO4 are needed, since typical impurities are catalytic.
Line 208. CNT is defined.
Line 284. “4.2 weight” missing “%”
Line 320 “inter-yard” means what?
Author Response

(The authors gave the same response as above.)

Reviewer 3 Report
This manuscript by Prof. Lobo et al. describes the investigation of the H2-stroage ability of various carbon-based materials. The results indicate that MWNT thread material has the highest wt % of H2 absorbed as high as 9 % compared to 6 %, 4 % and 2 % for graphene, SWNT network and MWNT network, respectively. It is indicated that the hydrogen storage capacity of carbon materials depends on surface area. Unfortunately, the data of BET surface area and pore volumes for all studied materials are unavailable. Is it a good idea to establish the relationship of the H2 capacity against the surface areas as well as pore volumes of the corresponding materials? These data will lend a big support to the authors’ statement albeit the N2-adsorption-desorption isotherm data are not able to directly reflect the H2 storage ability of the materials. The N2 isotherm would provide a positive trend for the capacity ability toward hydrogen molecules.Author Response
Green : Reviewer sugestions have been taken into account in the revised manuscript. Yellow: Comments and rebutal to the reviewer suggestions. Blue: Additional text included as responses to the reviewer suggestions.

Round 2
Reviewer 2 Report
The authors have responded to most of my comments. However, I strongly recommend that they include the galvanostatic charging curves in Supplementary Materials. Their response to this suggestion was that those results are not applicable to the main topic of the manuscript. Of course, I realized that when I made the suggestion to put them into SM and not the main text. Inclusion of such results would boost the confidence in the results for researchers experienced in this area and also be instructive for neophytes.
Author Response
Dear Editor
In the sequence of the reviewer recommendation regarding the inclusion of the galvanostatic charging curves, you can find them here in the Figure of Supplementary Material, with the aim of complementing the confidence in the results and not to make them the focus of the experimental work. These curves are displayed just with a fair quality because they result from a superposition of four paper recording plots.
We also take this opportunity to ask you for inclusion of just one more modification in the manuscript, because after the final revision we have made we noticed that in the last figure of the main text it is necessary to include the units in the xx axis (K-1).
Kind regards,
Rui Filipe Lobo
Corresponding author
